# Investigating the Relationship between CRISPR-Cas Content and Growth Rate in Bacteria

Zhi-Ling Liu,[a,b] En-Ze Hu,[a,b] Deng-Ke Niu[a,b]

[a]MOE Key Laboratory for Biodiversity Science and Ecological Engineering, College of Life Sciences, Beijing Normal University, Beijing, China
[b]Beijing Key Laboratory of Gene Resource and Molecular Development, College of Life Sciences, Beijing Normal University, Beijing, China

**ABSTRACT** CRISPR-Cas systems provide adaptive immunity for prokaryotic cells by recognizing and eliminating the recurrent genetic invaders whose sequences had been captured in a prior infection and stored in the CRISPR arrays as spacers. However, the biological/environmental factors determining the efficiency of this immune system have yet to be fully characterized. Recent studies in cultured bacteria showed that slowing the growth rate of bacterial cells could promote their acquisition of novel spacers. This study examined the relationship between the CRISPR-Cas content and the minimal doubling time across the bacteria and the archaea domains. Every completely sequenced genome could be used to predict a minimal doubling time. With a large data set of 4,142 bacterial samples, we found that the predicted minimal doubling times are positively correlated with spacer number and other parameters of the CRISPR-Cas systems, like array number, *Cas* gene cluster number, and *Cas* gene number. Different data sets gave different results. Weak results were obtained in analyzing bacterial empirical minimal doubling times and the archaea domain. Still, the conclusion of more spacers in slowly grown prokaryotes was supported. In addition, we found that the minimal doubling times are negatively correlated with the occurrence of prophages, and the spacer numbers per array are negatively associated with the number of prophages. These observations support the existence of an evolutionary trade-off between bacterial growth and adaptive defense against virulent phages.

**IMPORTANCE** Accumulating evidence indicates that slowing the growth of cultured bacteria could stimulate their CRISPR spacer acquisition. We observed a positive correlation between CRISPR-Cas content and cell cycle duration across the bacteria domain. This observation extends the physiological conclusion to an evolutionary one. In addition, the correlation provides evidence supporting the existence of a trade-off between bacterial growth/reproduction and antiviral resistance.

**KEYWORDS** CRISPR-Cas, minimal doubling time, bacteria, prophage, trade-off

CRISPR-Cas systems provide adaptive immunity by recognizing and eliminating the recurrent invaders whose genetic information had been captured in a prior infection and stored in the CRISPR arrays as spacers. Accumulating evidence indicates that slowing the host cell growth rate could enhance spacer acquisition, probably because the slow growth affects virus replication more than the cell's response to the invaders. Using a fluorescent CRISPR adaptation reporter method, Amlinger et al. (1) quantified the adaptation efficiency of type II CRISPR in different growth phases of *Escherichia coli*. They found that spacer acquisition occurs predominantly during the late exponential/early stationary phase. In line with this study, McKenzie et al. (2) observed that the *E. coli* cells that adapted and cleared the target DNA earlier grew significantly slower than the population means. The soil-dwelling bacterium *Pseudomonas aeruginosa* can infect plant and human bodies. It grows slowly in the cooler temperatures of the soil or during plant infection and rapidly under the higher temperature of human bodies.

Address correspondence to Deng-Ke Niu, dkniu@bnu.edu.cn.

The authors declare no conflict of interest.

Høyland-Kroghsbo et al. (3) found that decreasing the growth rate could increase the CRISPR adaptation frequency of the bacterium. Recently, Dimitriu et al. (4) found that the bacteriostatic agents that slow down the growth of *P. aeruginosa* could promote the phage-derived acquisition of novel spacers into the CRISPR array. Furthermore, they showed that limited carbon sources associated with slow bacterial growth could also enhance the evolution of CRISPR immunity. The latter result is in line with the previous observations of increased CRISPR-Cas immunity in bacteria grown in a nutrient-limited medium (5). Dormancy could be regarded as a prolonged growth state of the prokaryotic cells. In the response of the type VI CRISPR-Cas system to RNA virus infection, the Cas13 enzymes of the bacteria *Listeria ivanovii* were found to destruct both the RNA virus genomes and the host mRNA molecules, which further induce dormancy of the host bacterial cells (6). In the same study, the immunity of the bacterial cells against the unrelated virus was enhanced in the type VI CRISPR-Cas system-induced dormant state.

All the above results were obtained from analyzing a few cultured bacteria in laboratories. We could deduce that environmental factors that drive the bacteria to grow slowly in nature would enhance CRISPR-Cas immunity. The present study examines whether there is a universal relationship between growth rate and adaptive immunity across the bacteria domain. The null hypothesis to be tested is that CRISPR-Cas contents are unrelated to the growth rates across the bacteria and archaea domains.

## RESULTS

**Construction of the data sets.** We constructed three data sets of bacterial minimal doubling times and CRISPR-Cas system contents (Tables S1 to S3). The first data set contains 262 bacterial species with their empirical minimal doubling times retrieved from Madin et al. (7). Multiple strains have been sequenced for some bacterial species. In total, 424 genomes were downloaded from the GenBank database (8). These genomes have a wide variation in CRISPR-Cas contents (Table S4), on average, 1.34 CRISPR arrays (ranging from 0 to 12), 43.95 spacers (ranging from 0 to 665), 0.875 *Cas* gene clusters (ranging from 0 to 10), and 5.21 *Cas* genes (ranging from 0 to 49). Among them, 247 genomes do not have any CRISPR-Cas components, 20 have only CRISPR arrays or *Cas* gene clusters, and 157 have both CRISPR arrays and *Cas* gene clusters. For the species with multiple sequenced strains in these data set, we averaged the multiple CRISPR-Cas content values of each species for further analysis.

The second data set contains 4,142 bacterial genomes with their predicted minimal doubling times obtained from the EGGO database (accessed 26 November 2021) (9).

In the third data set, 508 bacterial genomes, each with only one CRISPR array and one *Cas* gene cluster, were selected from the second data set.

To control the effects of growth temperature, we also integrated the optimal growth temperatures predicted by the program Tome (version 1.0.0) (10) in the three data sets. Each species' predicted optimal growth temperatures in the first data set were also averaged.

From the descriptive statistics of the three data sets (Table 1), we can see that the predicted minimal doubling times have a similar median value and a similar range but a lower mean value than the empirical minimal doubling times. Phylogenetic generalized least-squares (PGLS) regression analysis showed that the empirical minimal doubling times and the predicted minimal doubling times are significantly correlated ($n = 252$, slope $= 0.228$, $P = 1.5 \times 10^{-10}$).

**Higher CRISPR-Cas contents in bacteria with long minimal doubling times.** We calculated the descriptive statistics for the minimal doubling times of the bacteria with the top 5% highest spacer numbers (Table 1). It appears that the bacteria rich in spacers grow slower than other bacteria. However, phylogenetic ANOVA analysis did not find significant differences between the top 5% CRISPR spacer-rich bacteria and the other 95% bacteria in the first or the third data set ($P > 0.05$ for both cases; Table 1). We suspected that their small sample sizes might account for the insignificant results.

Previously, Vieira-Silva and Rocha (11) discretized the minimum doubling time ($D$) into four classes: very fast ($D < 1$ h), fast ($1$ h $< D < 2$ h), intermediate ($2$ h $< D < 5$ h)

**TABLE 1** Descriptive statistics for the minimal doubling times of all the bacteria in our data sets and the top 5% CRISPR spacer-rich bacteria[a]

| Data sets | n | Median | Mean | SD | Maximum | Minimum |
|---|---|---|---|---|---|---|
| First data set | | | | | | |
| All bacteria | 262 | 2.66 | 6.71 | 8.70 | 52.80 | 0.23 |
| Top 5% | 13 | 6.50 | 11.57 | 10.62 | 36.94 | 0.92 |
| Second data set | | | | | | |
| All bacteria | 4,142 | 2.30 | 3.41 | 3.24 | 54.17 | 0.12 |
| Top 5% | 207 | 6.571 | 6.66 | 4.24 | 18.84 | 0.19 |
| Third data set | | | | | | |
| All bacteria | 508 | 2.25 | 3.60 | 4.16 | 54.17 | 0.14 |
| Top 5% | 25 | 2.88 | 4.51 | 4.85 | 19.99 | 0.58 |

[a]Please see the beginning of the Results section for the definition of the three data sets. We compared the minimal doubling times between the top 5% CRISPR spacer-rich bacteria and the other 95% bacteria in the three data sets using phylANOVA (12). The F values (and P values) obtained in analyzing the first, second, and third data sets are 4.3 (0.096), 231.3 (0.001), and 1.3 (0.356), respectively.

and slow ($D \geq 5$ h). Referring to their classification, our dichotomy between fast and slow was repeated three times, with one hour, two hours, and five hours as the boundary. We compared the slowly grown bacteria and fast-grown bacteria using the phylogenetic ANOVA method, phylANOVA (12), and found that the former had higher CRISPR-Cas contents, not just more spacers, than the latter in the second data set when slow was defined as $D \geq 5$ h and fast was defined as $D < 5$ h (Fig. 1A to D). Meanwhile, we noticed that the slowly grown bacteria have significantly higher optimal growth temperatures than the fast-grown bacteria (Fig. 1E). Significant differences were not found in the other two data sets or the phylogenetic ANOVA analyses using other definitions of fast and slow in the second data set. More results with significant differences were obtained when the CRISPR-Cas contents were compared using the Mann-Whitney $U$ test. As ignoring the effects of common ancestry in evolutionary analysis often gave false-positive results (13, 14), we are inclined to accept the results of the phylogenetic ANOVA analysis.

For a global relationship between CRISPR-Cas contents and growth rates across the bacterial domains, we performed PGLS regression analysis using CRISPR-Cas contents as the dependent variable and the minimal doubling times as the independent variable. The first data set has a positive correlation at a marginally significant level ($n = 262$, slope = 8.683, $P = 0.081$; Fig. 1F and Table 2) between the empirical minimal doubling times and the spacer numbers. To control the effects of growth temperatures, we performed multiple PGLS regression analyses using optimal growth temperatures as the second independent variable. In this analysis, the empirical minimal doubling times are significantly correlated with spacer numbers ($n = 262$, slope = 13.145, $P = 0.009$), $Cas$ gene cluster numbers ($n = 262$, slope = 0.166, $P = 0.035$), and $Cas$ gene numbers ($n = 262$, slope = 0.985, $P = 0.032$) (Table 2).

In the second data set, all the CRISPR-Cas content parameters (array number, spacer number, $Cas$ gene cluster number, and $Cas$ gene number) have significant positive correlations with the predicted minimal doubling times, no matter whether the optimal growth temperatures were controlled or not (slope > 0, $P < 0.05$ for all cases; Table 2).

The increase of spacer number with the minimal doubling time can be explained by an enhancing effect on spacer acquisition of slow growth, as previously observed in the cultured bacteria (1–6). However, slowly grown bacteria may also have more spacers because they have a more significant number of CRISPR arrays, either duplicated or acquired horizontally. The same logic exists for the $Cas$ gene number. Therefore, we examined the relationship of minimal doubling times with the spacer number per array and $Cas$ gene number per cluster. Here, only genomes with arrays or $Cas$ gene clusters were counted. Genomes without CRISPR-Cas systems were not included in the denominator. The spacer numbers per array positively correlate with the empirical minimal

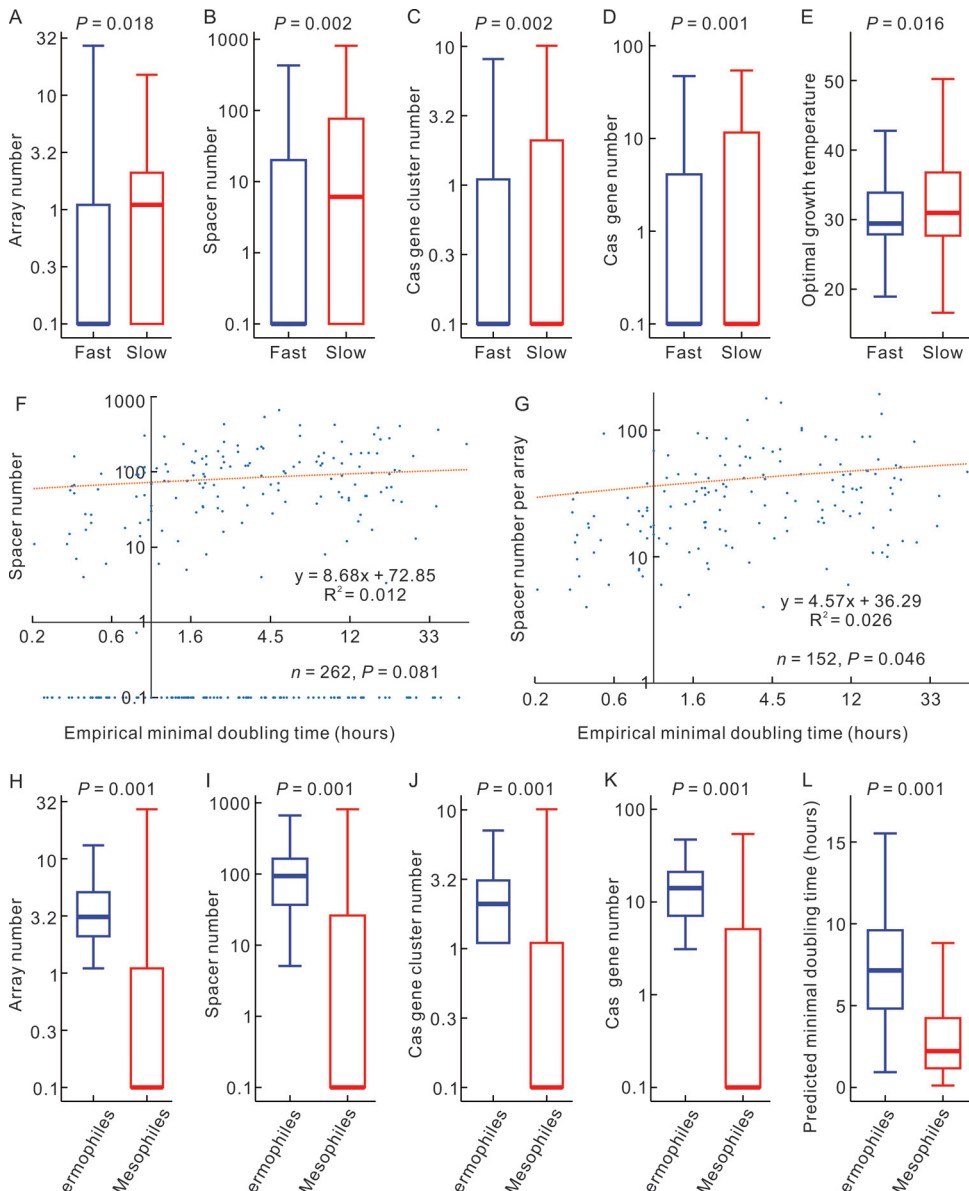

**FIG 1** Correlations among bacterial CRISPR-Cas contents, the minimal doubling times, and optimal growth temperatures. (A to E) Boxplots show higher CRISPR-Cas contents and higher optimal growth temperatures in slowly grown bacteria ($n = 891$) than in fast-grown bacteria ($n = 3,251$). The slowly grown was defined as the minimal doubling times $\geq$ 5h and the fast-grown was defined as the minimal doubling times <5 h. The comparisons were performed using phylANOVA with the $P$ values adjusted by the Benjamini-Hochberg (BH) procedure, and the F values of panels A to E are 178.0, 293.5, 341.4, 365.7, and 186.6, respectively. (F) The CRISPR spacer numbers and (G) spacer number per array in bacteria are positively correlated with the empirical minimal doubling times. The correlations were examined using PGLS regression analyses, with the positive sign of the regression slope to indicate the positive correlation. Actual hours were labeled in these two panels just because they are intuitive. The doubling time values were log-transformed using base $e$ in the PGLS analysis. (H to L) Boxplots show higher CRISPR-Cas contents and longer doubling times in thermophilic bacteria ($n = 156$) than in mesophilic bacteria ($n = 3,944$). The comparisons were performed using phylANOVA with the $P$ values adjusted by the Benjamini-Hochberg (BH) procedure, and the F values of panels H to L are 321.6, 474.9, 509.0, 634.0, and 241.8, respectively. Following reference (38), mesophiles were defined as $20 \leq$ optimal growth temperature $< 45°C$, and thermophiles were defined as $45 \leq$ optimal growth temperature $< 80°C$.

doubling times ($n = 152$, slope = 4.570, $P = 0.046$; Fig. 1G) and the predicted minimal doubling times ($n = 1,507$, slope = 3.282, $P = 0.010$). However, the *Cas* gene numbers per cluster are not significantly correlated with the empirical minimal doubling time ($n = 149$, slope = $-0.003$, $P = 0.977$) or the predicted minimal doubling time ($n = 1,472$,

**TABLE 2** Correlations of the minimal doubling times (*D*) with CRISPR-Cas contents[a]

| Data sets | N | D | | D with OPT controlled | | OPT with D controlled | |
|---|---|---|---|---|---|---|---|
| | | Slope | P | Slope | P | Slope | P |
| **First data set** | | | | | | | |
| Array no. | 262 | 0.035 | 0.791 | 0.138 | 0.285 | **0.056** | **$4 \times 10^{-5}$** |
| Spacer no. | 262 | 8.683 | 0.081 | **13.145** | **0.009** | **2.187** | **$4 \times 10^{-4}$** |
| Spacer no. per array | 152 | **4.570** | **0.046** | **5.047** | **0.033** | 0.183 | 0.439 |
| *Cas* gene cluster no. | 262 | 0.096 | 0.235 | **0.166** | **0.035** | **0.039** | **$3 \times 10^{-6}$** |
| *Cas* gene no. | 262 | 0.505 | 0.286 | **0.985** | **0.032** | **0.261** | **$10^{-7}$** |
| *Cas* gene no. per cluster | 149 | −0.003 | 0.977 | 0.066 | 0.552 | **0.025** | **0.010** |
| **Second data set** | | | | | | | |
| Array no. | 4,142 | **0.168** | **0.002** | **0.151** | **0.005** | **0.056** | **$10^{-14}$** |
| Spacer no. | 4,142 | **5.326** | **0.003** | **5.212** | **0.003** | **1.456** | **$2 \times 10^{-8}$** |
| Spacer no. per array | 1,507 | **3.282** | **0.010** | **3.408** | **0.008** | −0.101 | 0.434 |
| *Cas* gene cluster no. | 4,142 | **0.091** | **0.001** | **0.085** | **0.002** | **0.033** | **$10^{-17}$** |
| *Cas* gene no. | 4,142 | **0.562** | **$9 \times 10^{-4}$** | **0.528** | **0.001** | **0.203** | **$3 \times 10^{-18}$** |
| *Cas* gene no. per cluster | 1,077 | 0.106 | 0.145 | 0.105 | 0.153 | $6 \times 10^{-4}$ | 0.931 |
| **Third dataset** | | | | | | | |
| Spacer no. | 508 | **4.351** | **0.029** | **4.588** | **0.021** | −0.421 | 0.115 |
| *Cas* gene no. | 508 | 0.177 | 0.107 | 0.172 | 0.118 | 0.010 | 0.515 |

[a]OPT, optimal growth temperature. Please see the beginning of the Results section for the definition of the data sets. The correlations were examined using multiple PGLS regression analyses, with the positive sign of the regression slope to indicate the positive correlation. Significant, marginally significant, and nonsignificant results were shown in bold, regular, and gray font, respectively.

slope = 0.106, *P* = 0.145). In addition, we also examined the relationship of predicted minimal doubling times with spacer numbers and *Cas* gene numbers in the third data set, i.e., 508 bacterial genomes, each of which has only one CRISPR array and one *Cas* gene cluster. Only spacer numbers significantly correlate with the predicted minimal doubling times, regardless of whether optimal growth temperatures were controlled (Table 2).

According to the *Cas* genes (15), 508 bacteria in the third data set could be classified, including 329 bacteria with the class 1 *Cas* gene cluster (304 type I, 21 type III, and four type IV) and 173 bacteria with the class 2 *Cas* gene cluster (167 type II, 5 type V, and 1 type VI). We performed the PGLS regression analyses for the groups with more than 50 samples, class 1, class 2, type I, and type II. Although the bacteria with the type I *Cas* gene cluster is a subset of the bacteria with the class 1 *Cas* gene cluster, they did not give consistent correlations. In bacteria with the class 1 *Cas* gene cluster, the spacer numbers are marginally correlated with the minimal doubling times after controlling the optimal growth temperatures (*n* = 329, slope = 4.028, *P* = 0.097). However, the relationship is far from statistically significant in the bacteria with the type I *Cas* gene cluster (*P* > 0.10). Significant (*P* < 0.05) or marginally significant (0.05 ≤ *P* < 0.1) correlations have not been observed in bacteria with class 2 or type II *Cas* gene clusters. Different types of CRISPR-Cas systems seem not identical in their relationships with minimal doubling times. Future studies with larger samples will be helpful for a conclusion.

**CRISPR-Cas content and growth temperature.** Previous studies observed that bacterial CRISPR abundance is positively correlated with growth temperatures (16–19), with an abrupt jump at around 45°C (20). Here, we first compared the CRISPR-Cas contents and predicted minimal doubling times between thermophilic and mesophilic bacteria using the phylogenetic ANOVA method, phylANOVA (12). As shown in Fig. 1H to L, thermophilic bacteria have higher CRISPR-Cas contents and longer doubling times. Therefore, we examined the relationship between the CRISPR-Cas contents and optimal growth temperatures by controlling the minimal doubling time using multiple PGLS regression analyses. The results show that the minimal doubling times and the growth temperatures are independently correlated with CRISPR-Cas contents (Table 2).

**TABLE 3** Correlations between the prophage contents and CRISPR-Cas contents[a]

| Datasets | N | Prophage no. | | Prophage existence | |
|---|---|---|---|---|---|
| | | Slope | P | Slope | P |
| First data set | | | | | |
| Array no. | 262 | −0.025 | 0.606 | 0.003 | 0.783 |
| Spacer no. | 262 | **−0.002** | **0.037** | $-3 \times 10^{-4}$ | 0.313 |
| Spacer no. per array | 262 | **−0.010** | **0.007** | −0.001 | 0.268 |
| *Cas* gene cluster no. | 262 | −0.062 | 0.422 | −0.010 | 0.633 |
| *Cas* gene no. | 262 | −0.013 | 0.332 | −0.003 | 0.321 |
| *Cas* gene no. per cluster | 262 | −0.085 | 0.316 | **−0.054** | **0.029** |
| | | | | | |
| Second data set | | | | | |
| Array no. | 4,142 | 0.017 | 0.256 | −0.002 | 0.621 |
| Spacer no. | 4,142 | $-3 \times 10^{-4}$ | 0.421 | $-10^{-4}$ | 0.216 |
| Spacer no. per array | 4,142 | **−0.005** | **0.002** | −0.001 | 0.160 |
| *Cas* gene cluster no. | 4,142 | 0.039 | 0.169 | −0.001 | 0.856 |
| *Cas* gene no. | 4,142 | 0.005 | 0.341 | −0.001 | 0.608 |
| *Cas* gene no. per cluster | 4,142 | −0.028 | 0.284 | −0.005 | 0.448 |
| | | | | | |
| Third data set | | | | | |
| Spacer no. | 508 | **−0.005** | **0.009** | **−0.001** | **0.093** |
| *Cas* gene no. | 508 | −0.016 | 0.672 | 0.002 | 0.837 |

[a]The correlations were examined using PGLS regression analyses, with the negative sign of the regression slope to indicate the negative correlation. The statistically significant (and marginally significant) cases are shown in bold. Prophage existence: 0 and 1 were designated absence and presence in the PGLS analysis. Please see the beginning of the Results section for the definition of the three data sets.

However, the spacer numbers per array and the *Cas* gene numbers per cluster in the second data set are not significantly correlated with the optimal growth temperatures ($P > 0.10$ for both cases; Table 2). Similarly, the spacer and *Cas* gene numbers in the third data set are not significantly correlated with the optimal growth temperatures ($P > 0.10$ for both cases; Table 2). It seems that the increase of adaptive immunity with growth temperature is mainly achieved by increasing the number of CRISPR arrays and *Cas* gene clusters. In contrast, increasing adaptive immunity with minimal doubling time seems achieved by increasing all aspects of the CRISPR-Cas systems.

**Relationship between prophage contents, minimal doubling times, and CRISPR-Cas contents.** Previously, Touchon et al. (21) found that the frequency of prophage is negatively correlated with the minimal doubling time in bacteria. We examined this relationship using our data sets. Neither prophage numbers nor the presence/absence of prophages significantly correlate with bacterial empirical minimal doubling times ($P > 0.10$ for both cases). However, both prophage numbers and the presence/absence of prophages are negatively correlated with bacterial predicted minimal doubling times ($n = 4,142$, slope $= -0.186$, $P = 5 \times 10^{-5}$ and $n = 4,142$, slope $= -0.057$, $P = 10^{-5}$, respectively). In the third data set, the presence/absence of prophages is negatively correlated with the predicted minimal doubling time ($n = 508$, slope $= -0.052$, $P = 0.040$). Furthermore, we examined the relationship between prophage contents and CRISPR-Cas systems. The spacer numbers per array in the first and second data sets and the spacer numbers in the third data set consistently correlate negatively with the prophage numbers ($P < 0.05$ for all the cases; Table 3). However, only the spacer numbers of the third data set are negatively correlated (at a marginally significant level) with the prophage existence. Table 3 shows that prophages have weak or no relationship with the array numbers, the *Cas* gene cluster numbers, or the *Cas* gene numbers.

## DISCUSSION

In a few cultured bacterial species, slow growth has been found to enhance spacer acquisition (1–6). Our phylogenetic analyses showed that the spacer numbers and the other CRISPR-Cas content parameters (array number, *Cas* gene cluster number, and *Cas* gene number) positively correlate with the minimal doubling times in bacteria.

Previous authors reporting the enhancing effects of slow growth on spacer acquisition generally explain their observation by the mechanisms of spacer acquisition (1, 3, 4). If the slow growth could enhance spacer acquisition, the CRISPR-Cas systems would be an efficient mechanism to counteract viruses in slowly grown cells, and to be selected when the growth rate has been slowed down on an evolutionary scale.

Furthermore, we could put the CRISPR-Cas-enhancing effect of slow growth into an ecological context. Studies on natural bacterial communities support the existence of trade-offs between intraspecific competition ability (faster growth) and invulnerability to predation (including virus resistance) (22–25). With the limited resources available in natural environments, the bacteria that allocate more resources to defense must grow slowly, and the bacteria with high competition ability allocate less amounts of resources to virus resistance. The trade-off between faster growth and defense is destined if maintaining and expressing the CRISPR-Cas system exert a significant metabolic burden. This metabolic burden has been observed in *Streptococcus thermophilus* by comparing a strain that constitutively expresses the Cas protein and a strain with a nonfunctional mutant of Cas (26). However, acquiring a few spacers was not associated with fitness costs (26). Besides the metabolic burden, autoimmunity induced by self-and prophage-targeting spacers is another widely cited cost of CRISPR-Cas systems (27, 28). A trade-off between fast growth and defense using CRISPR-Cas systems might be mediated by the autoimmunity targeting prophages. In the present study, we found evidence for the mutually exclusive effects between prophage and CRISPR-Cas immunity across the bacterial domain (Table 3). Because prophages are more frequent in bacteria with short minimal doubling times (21), there should be a selective force against CRISPR–Cas systems in fast-grown bacteria (29, 30).

In this study, the three bacterial data sets did not always give consistent results. We suggest that the differences in the results should be attributed to the differences in the accuracy and the data sets' sample size. In a previous study, we emphasized that phylogenetic comparative studies generally required larger samples than typical statistical analyses (31). Here, we want to briefly discuss the empirical and predicted data's advantages and disadvantages. At first glance, the empirical minimal doubling times seem superior to the predicted minimal doubling times because the prediction methods could not achieve an accuracy of 100%. As the wide-spread diversity among the strains designated one bacterial species (32), the ideal empirical data should include different parameters (e.g., minimal doubling time, optimal growth temperature, and CRISPR-Cas contents) collected from the same strain. Unfortunately, this is not always the case. In many cases, a phenotype value is assigned to a bacterial species without mentioning the strain identity, so the values of different phenotypes might come from different strains. In contrast, all the predicted phenotype values of a strain come from the genome sequences of the same strain. Potential errors resulting from polymorphism within each species could be eliminated if all the parameters were predicted/annotated from genome sequences.

We also examined the relationships in archaea using PGLS regression. First, we found that the predicted minimal doubling times are not significantly correlated with the empirical minimal doubling times ($n = 100$, slope = 0.055, $P = 0.325$). The method to predict minimal doubling times was mainly based on bacterial data (9). It seems not to work accurately in archaea. Therefore, we only examined the relationship between archaea's empirical minimal doubling times and CRISPR-Cas contents. Univariate PGLS regression did not find significant correlations ($P > 0.10$ for all the cases). However, after controlling optimal growth temperatures, the empirical minimal doubling times are significantly correlated with the array numbers ($n = 103$, slope = 0.838, $P = 0.027$), spacer numbers ($n = 103$, slope = 23.239, $P = 0.022$), *Cas* gene cluster numbers ($n = 103$, slope = 0.378, $P = 0.005$), and *Cas* gene numbers ($n = 103$, slope = 2.120, $P = 0.017$). Meanwhile, the spacer numbers per array positively correlate with the empirical minimal doubling time ($n = 88$, slope = 7.476, $P = 0.018$). Because of the limited number of samples, we did not analyze the archaea in such detail as the bacteria.

## MATERIALS AND METHODS

The data sets of prokaryotic empirical minimal doubling times and predicted minimal doubling times were retrieved from reference (7) and the EGGO database (accessed 26 November 2021) (9), respectively. To avoid the outlier effects on the statistical results, we only retained the minimal doubling time shorter than 60 h. Because the data on minimal doubling times are highly skewed, they were log-transformed using base $e$. To cover more prokaryotes, we used the optimal growth temperatures predicted by the program Tome (version 1.0.0) (10). The phylogenetic trees and taxonomic information of bacteria and archaea were retrieved from the Genome Taxonomy Database (GTDB; accessed 8 April 2022) (33). Because each prokaryotic species rather than each strain has one empirical minimal doubling time value, phylogenetic comparative analysis of the empirical minimal doubling times requires a species tree. From the GTDB, we constructed a species tree using the representative genome of each species. The prokaryotic genome assembly files were downloaded from the GenBank database (accessed 26 April 2021) (8). The CRISPR-Cas systems of the prokaryotic genomes were annotated using CRISPRCasFinder (version 4.2.20) (34). Using VirSorter 2.2.3 (35), prophages were found in bacteria but not archaeal genomes. To avoid underestimation of the CRISPR-Cas systems, partially assembled genomes have been discarded, and only those with assembly levels of "complete genome" or "chromosome" were retained in our analyses. The *Cas* gene number of each type of *Cas* gene cluster is relatively constant in evolution, but different types and classes have a significant difference in their *Cas* gene number, ranging from 1 to 11 (15). The difference in *Cas* gene number between genomes should be attributed to different types of *Cas* gene clusters.

We estimated the phylogenetic signals ($\lambda$) of the analyzed characters using the R (version 4.0.2) package phytools (version 0.7-47) (36). All the analyzed characters exhibit significant phylogenetic signals, indicating that phylogenetic comparative methods are required in the analyses to control the effect of common ancestry. The correlations between different characters were estimated by the PGLS regression using the R (version 4.0.2) package phylolm (version 2.6.2) (37). Pagel's $\lambda$ model has been applied in this study. The phylogenetic ANOVA analyses were performed using phylANOVA (12) with the *P* values adjusted by the Benjamini-Hochberg (BH) procedure.

## SUPPLEMENTAL MATERIAL

Supplemental material is available online only.

**SUPPLEMENTAL FILE 1**, XLSX file, 0.6 MB.

## ACKNOWLEDGMENTS

This work was supported by the National Natural Science Foundation of China (grant number 31671321). We thank the anonymous reviewers for their constructive comments.

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
