## [Reviewer comments · Microbiology Spectrum]

Microbiology Spectrum

Investigating the relationship between CRISPR-Cas content and growth rate in bacteria

Zhi-Ling Liu, En-Ze Hu, and Deng-Ke Niu

Corresponding Author(s): Deng-Ke Niu, Beijing Normal University

Review Timeline:

Submission Date:	August 26, 2022
Editorial Decision:	October 31, 2022
Revision Received:	February 15, 2023
Editorial Decision:	February 27, 2023
Revision Received:	March 7, 2023
Accepted:	March 11, 2023

Editor: David Pride

Reviewer(s): Disclosure of reviewer identity is with reference to reviewer comments included in decision letter(s). The following individuals involved in review of your submission have agreed to reveal their identity: Kaan Çeylan (Reviewer #2)

Transaction Report:

DOI: <https://doi.org/10.1128/spectrum.03409-22>

October 31, 2022

Dr. Deng-Ke Niu
College of Life Sciences, Beijing Normal University
Beijing
China

Re: Spectrum03409-22 (A Positive Correlation between CRISPR Spacer Abundance and Cell Cycle Duration Across the Bacteria Domain)

Dear Dr. Deng-Ke Niu:

Link Not Available

Sincerely,

David Pride

Journals Department
Reviewer comments:

Reviewer #1 (Comments for the Author):

Liu et al. present a study of correlation between CRISPR spacer abundance and cell cycle duration in bacteria. Specifically, the authors present a table reporting a correlation between CRISPR spacer abundance and cell cycle duration in bacteria, with statistical significance. Though this is generally an interesting topic and an intriguing finding, there are several issues that need to be addressed in order to assess the reported findings, regarding methods, data, results, analyses and conclusions.

As the authors point out, several recent studies have shown that slowing down the host growth rate during CRISPR spacer acquisition enables more efficient capture of protospacers. A finding in select systems in which active CRISPR-Cas systems are in action is one thing, but I am unsure this indicates that fast growing bacteria have shorter arrays, or that slow growing bacteria

have long arrays. Rather, slowing down growth during acquisition enables more time for capture and spacer integration. I suggest the authors define a hypothesis: (H0: growth rate has no effect on spacer size) and test it (hopefully disprove it).

The authors state that they "examined whether there is a universal relationship between growth rate and adaptive immunity across the bacteria domain". This should start with an analysis of CRISPR occurrence in the domain in question (do CRISPR-Cas system occur more frequently in slow growing bacteria?), and then determine whether there is a relationship between CRISPR array size and growth rate. This may also need to take into account the occurrence (or not) of multiple CRISPR-Cas systems in bacteria (when more than one system is present, is there a relationship between number of systems and growth rate?). Split arrays should also be taken into account if and where applicable.

In perhaps the most interesting and valuable dataset, the authors focus on 262 bacteria with empirical doubling time. It is not clear how many of these have CRISPR-Cas immune systems, how many they have, and what the range of spacer numbers are. Also, several studies have shown that spacer number can vary significantly within a given species and sometimes, within a specific array occurring in a species (let alone the variable occurrence of some arrays in select species). How do the authors account for patterns of variable occurrence of CRISPR-Cas systems within a species, and the variable size of the array (when a system is present) within a species? This indicates that the same growth rate could be applicable to widely different array sizes...

A list of these 262 bacteria and their systems and their sizes and their variability in occurrence and size and their documented activity would be desirable, if not necessary. For select species - systems combinations that have been documented, does the correlation apply? The methods do not explain how spacer abundance was determined. This must be made available and to some extent validated (especially for the aforementioned 262 dataset).

It is unclear how reliable the predicted minimal doubling time predictions are. Without some background on the predictions, it is difficult, if not impossible, to assess some of the data presented in the paper. Since this is a significant contributor to the data presented, this must be somewhat validated.

There is no information on the CRISPR loci used in the study. It is not clear how many systems, how many loci, and how spacer abundance was calculated or determined, especially in cases where multiple systems and multiple loci occur in a genome, including in cases where CRISPR loci are in trans relative to the cas operon.

Overall, the study is relatively preliminary (it is a short paper with a single table for results), and many details are missing, as well as complete analyses focusing on important details (number of spacers, types of CRISPR-Cas systems, number of CRISPR arrays). The study would benefit from additional analyses investigating these aspects/

While CRISPR-Cas systems occur in ~40% of bacteria, they are encoded in the large majority (~90%) of archaea. The finding that no correlation is observed between spacer abundance and growth rate in archaea is a major concern given the disagreement with the major conclusion of the study. This needs to be discussed and explained.

The authors conclude that their observation provides direct evidence supporting the existence of trade-offs between cell growth rate and viral resistance in a "killing the winner hypothesis". I am unsure that the data provided in the paper supports the conclusion, let alone that it provides direct evidence in support of its existence. The authors mention that slow growth affects virus replication, which may be partially distinct from mechanisms that provide resistance at different time scales. Determining the trade offs in question would help clarify and possibly support this strong conclusive statement. The authors discuss prophages in the closing section of the manuscript. The study should determine whether indeed there is a correlation between the occurrence of CRISPR-Cas systems and prophages and whether spacer abundance correlates with the absence of prophages.

For the N=262 dataset, the authors should provide some graphical display of the correlation in question and also provide some descriptive statistics of the entries. Specific analyses should be carried out by CRISPR-Cas class, type and possibly subtype to determine correlations (especially the most abundant class 1; type I and III; and I-B, I-C, I-E, respectively).

In some cases, especially cases where CRISPR loci are unusually large, the authors should determine what the doubling time is for the host and whether there is a correlation.

Reviewer #2 (Comments for the Author):

Dear author,
I do not have any additional suggestions regarding your work.
I wish good work.

Staff Comments:

Preparing Revision Guidelines

Please return the manuscript within 60 days; if you cannot complete the modification within this time period, please contact me. If you do not wish to modify the manuscript and prefer to submit it to another journal, please notify me of your decision immediately so that the manuscript may be formally withdrawn from consideration by Microbiology Spectrum.

Reviewer #1 (Comments for the Author):

Liu et al. present a study of correlation between CRISPR spacer abundance and cell cycle duration in bacteria. Specifically, the authors present a table reporting a correlation between CRISPR spacer abundance and cell cycle duration in bacteria, with statistical significance. Though this is generally an interesting topic and an intriguing finding, there are several issues that need to be addressed in order to assess the reported findings, regarding methods, data, results, analyses and conclusions.

Authors' response:

We thank the reviewer for the detailed, constructive comments. Revision according to these comments greatly expanded this manuscript, changing it from a short Observations paper to a Research Article. We believe that this change will benefit the readers.

As the authors point out, several recent studies have shown that slowing down the host growth rate during CRISPR spacer acquisition enables more efficient capture of protospacers. A finding in select systems in which active CRISPR-Cas systems are in action is one thing, but I am unsure this indicates that fast growing bacteria have shorter arrays, or that slow growing bacteria have long arrays. Rather, slowing down growth during acquisition enables more time for capture and spacer integration. I suggest the authors define a hypothesis: (H0: growth rate has no effect on spacer size) and test it (hopefully disprove it).

Authors' response:

We suspect the "spacer size" in the above comment is a typo. According to the context, the H0 should be "growth rate has no effect on spacer number (or array size)." We explicitly defined it in the revised manuscript. Please see lines 73-74.

The authors state that they "examined whether there is a universal relationship between growth rate and adaptive immunity across the bacteria domain". This should start with an analysis of CRISPR occurrence in the domain in question (do CRISPR-Cas system occur more frequently in slow growing bacteria?), and then determine whether there is a relationship between CRISPR array size and growth rate. This may also need to take into account the occurrence (or not) of multiple CRISPR-Cas systems in bacteria (when more than one system is present, is there a relationship between number of systems and growth rate?). Split arrays should also be taken into account if and where applicable.

Authors' response:

Thanks for these detailed comments.

In the revised manuscript, we divided the bacteria into two groups according to their growth rates and found that bacteria with $D \geq 5h$ have significantly higher CRISPR-Cas contents than fast-grown bacteria (please see lines 104-111 and Table 2).

To control the effect of multiple CRISPR-Cas systems in one bacterial genome, we added two parameters, Spacer numbers per array and Cas gene numbers per cluster, and constructed a dataset including the bacterial genomes, each with only one CRISPR array and one Cas gene

cluster (Please see lines 90-91 and Table S3).

In perhaps the most interesting and valuable dataset, the authors focus on 262 bacteria with empirical doubling time. It is not clear how many of these have CRISPR-Cas immune systems, how many they have, and what the range of spacer numbers are. Also, several studies have shown that space number can vary significantly within a given species and sometimes, within a specific array occurring in a species (let alone the variable occurrence of some arrays in select species). How do the authors account for patterns of variable occurrence of CRISPR-Cas systems within a species, and the variable size of the array (when a system is present) within a species? This indicates that the same growth rate could be applicable to widely different array sizes...

Authors' response:

In the first dataset, there is only one empirical minimal doubling time for a bacterial species. Multiple strains have been sequenced for most bacterial species, so each species have multiple CRISPR-Cas content values. We briefly described the CRISPR-Cas contents and variations of these genomes, then averaged each species' multiple CRISPR-Cas content values. Because each genome has its own predicted minimal doubling time and CRISPR-Cas content value, the within-species variations do not affect the results of the second and third datasets (Please see lines 77-87 and 235-250).

A list of these 262 bacteria and their systems and their sizes and their variability in occurrence

and size and their documented activity would be desirable, if not necessary. For select species - systems combinations that have been documented, does the correlation apply? The methods do not explain how spacer abundance was determined. This must be made available and to some extent validated (especially for the aforementioned 262 dataset).

Authors' response:

We changed the words "spacer abundance" in the revised manuscript to "spacer number." An excel file containing the CRISPR-Cas contents and the minimal doubling times of the three datasets is submitted as online supplementary data (Table S1-S4).

It is unclear how reliable the predicted minimal doubling time predictions are. Without some background on the predictions, it is difficult, if not impossible, to assess some of the data presented in the paper. Since this is a significant contributor to the data presented, this must be somewhat validated.

Authors' response:

A correlation was calculated by the authors of the predicted minimal doubling times. In the revised manuscript, we also performed a PGLS regression analysis showing a significant correlation between the predicted minimal doubling times and the empirical minimal doubling times (please see lines 95-100).

There is no information on the CRISPR loci used in the study. It is not clear how many systems,

how many loci, and how spacer abundance was calculated or determined, especially in cases where multiple systems and multiple loci occur in a genome, including in cases where CRISPR loci are in trans relative to the cas operon.

Overall, the study is relatively preliminary (it is a short paper with a single table for results), and many details are missing, as well as complete analyses focusing on important details (number of spacers, types of CRISPR-Cas systems, number of CRISPR arrays). The study would benefit from additional analyses investigating these aspects/

Authors' response:

Thanks for this constructive comment. We changed the title of the manuscript from "spacer abundance" to "CRISPR-Cas contents" to include four parameters: Array number, Spacer number, Cas gene cluster number, and Cas gene number. The results showed that the spacer number and the other three parameters positively correlate with the minimal doubling times. Besides extending the laboratory results of a few cultured bacteria into the whole bacteria domain, the revised manuscript showed that other aspects of the CRISPR-Cas systems are also enhanced in the slowly-grown bacteria.

While CRISPR-Cas systems occur in ~40% of bacteria, they are encoded in the large majority (~90%) of archaea. The finding that no correlation is observed between spacer abundance and growth rate in archaea is a major concern given the disagreement with the major conclusion of the study. This needs to be discussed and explained.

Authors' response:

In the revised analysis, although we still did not find a positive correlation between minimal doubling times and spacer numbers, we found that spacer number per array is positively correlated with minimal doubling time in archaea. In addition, the CRISPR-Cas contents are correlated with minimal doubling times after controlling the optimal growth temperatures. (Please see lines 177-188).

The authors conclude that their observation provides direct evidence supporting the existence of trade-offs between cell growth rate and viral resistance in a "killing the winner hypothesis". I am unsure that the data provided in the paper supports the conclusion, let alone that it provides direct evidence in support of its existence. The authors mention that slow growth affects virus replication, which may be partially distinct from mechanisms that provide resistance at different time scales. Determining the trade offs in question would help clarify and possibly support this strong conclusive statement.

Authors' response:

Thanks for this comment. We have toned down the conclusion of the revised manuscript. In addition, more discussions on the trade-offs have been added (Please see lines 217-234).

The authors discuss prophages in the closing section of the manuscript. The study should determine whether indeed there is a correlation between the occurrence of CRISPR-Cas systems

and prophages and whether spacer abundance correlates with the absence of prophages.

Authors' response:

Thanks for this constructive comment. In the revised manuscript, we first examined the previously reported pattern of more prophages in bacteria with small minimal doubling times using our datasets, then performed the analyses suggested by the reviewer (Please see lines 189-205 and Table 4).

For the N=262 dataset, the authors should provide some graphical display of the correlation in question and also provide some descriptive statistics of the entries. Specific analyses should be carried out by CRISPR-Cas class, type and possibly subtype to determine correlations (especially the most abundant class 1; type I and III; and I-B, I-C, I-E, respectively).

In some cases, especially cases where CRISPR loci are unusually large, the authors should determine what the doubling time is for the host and whether there is a correlation.

Authors' response:

The descriptive statistics for the empirical minimal doubling times of all the bacteria and the bacteria with unusually large CRISPR loci (top 5%) were shown in Table 1 of the revised manuscript. A pattern could be seen that spacer-rich bacteria have longer doubling times.

In the revised manuscript, a figure has been added to show the relationship of the empirical minimal doubling times with spacer number and spacer number per array (Please see Fig. 1).

In the genomes with multiple CRISPR-Cas systems, the CRISPR loci may act *in trans* with the *Cas* operon. Therefore, we can only precisely study the correlations in different types or classes in genomes with only one CRISPR array and one *Cas* cluster. We found only 16 bacteria from the first dataset, of which 11 were with type I *Cas* loci and five were with type II *Cas* loci. The sample sizes of all these types and classes are too small to perform phylogenetic comparative analyses. We analyzed the correlations of different CRISPR-Cas types and classes using the predicted minimal doubling times in the third dataset as an alternative. The results have been shown in lines 149-162.

Reviewer #2 (Comments for the Author):

Dear author,

I do not have any additional suggestions regarding your work.

I wish good work.

Authors' response:

Thanks for your review.

February 27, 2023

Dr. Deng-Ke Niu
Beijing Normal University
College of Life Sciences
Beijing 100875
China

Re: Spectrum03409-22R1 (A Positive Correlation between CRISPR-Cas Contents and Cell Cycle Durations Across the Bacteria Domain)

Dear Dr. Deng-Ke Niu:

Link Not Available

Sincerely,

David Pride

Journals Department
Reviewer comments:

Reviewer #1 (Comments for the Author):

Liu et al. present a revised version of their study of correlation between CRISPR spacer abundance and cell cycle duration in bacteria. The rebuttal and revisions are noted across the board and substantially improved the manuscript. A few issues should still be addressed, as outlined below.

The title might still need some work, and by foregoing archaea and sticking to bacteria, perhaps the authors could consider or adapt the following: "investigating the relationship between CRISPR-Cas content and growth rate in bacteria".

As the authors state in their rebuttal, they have divided bacteria into two groups according to growth rate. I think they should consider showcasing this split to display the data in figure 1 (possibly different colors in the same panel, or different panels) and add a bar graph visually showing growth rate between these two groups and determining statistical significance (besides table 2).

Given the variation in cas gene number inherent to the CRISPR-Cas system nomenclature, the authors may elect to either keep the data and add a disclosure to that effect, or forego investigating cas gene number. Both may be agreeable, but the choice should be justified explicitly.

As the authors mention, temperature plays an important part in host biology. I also know that thermophilic organisms have been discussed to have "interesting" CRISPR-Cas patterns. The authors may elect to compare and contrast growth rate and CRISPR-Cas patterns between mesophiles and thermophiles, if and where applicable (a refined figure display, or options with different color usage).

As mentioned initially, the conflicting results from archaea are noteworthy and concerning. I suggest the authors focus on bacteria only in their study and then discuss archaea in the discussion section (but keep some of the data in the tables, perhaps not the figure), to contextualized this and also keep the most significant results and narrative in the paper. This is phylogenetically defensible, and if presented correctly, agreeable and understandable for the readership. "Only" bacteria are called out in the title afterall...

Figure 1 is a welcome addition, but needs improvements. The authors should provide several panels (see comment above), including bar graphs comparing and contrasting subsets of organisms and patterns (e.g. thermophilic vs mesophilic, fast growing vs slow growing, and possibly more). Visually, the authors should consider using a log scale for the y axis (log 10 scale might be appropriate given the numbers in play) and also a clearer label and scale for the x axis (not intuitive for microbiologists). Actual hours should be labelled, even if the scale is log-transformed. Also, the authors should consider adding relevant numbers (p-values) to the graph, and consider adding the R-square (on the figure, or in the legend, as the readership would assuredly wonder).

Where applicable, statistical analyses must be performed to show significance (or not) of the data (e.g. Table 1, though tables 2 and 3 are noted).

Grammatically and stylistically, the paper would benefit from copy-editing, including for the table entries.

Staff Comments:

Preparing Revision Guidelines

Please return the manuscript within 60 days; if you cannot complete the modification within this time period, please contact me. If you do not wish to modify the manuscript and prefer to submit it to another journal, please notify me of your decision immediately so that the manuscript may be formally withdrawn from consideration by Microbiology Spectrum.

Reviewer #1 (Comments for the Author):

Liu et al. present a revised version of their study of correlation between CRISPR spacer abundance and cell cycle duration in bacteria. The rebuttal and revisions are noted across the board and substantially improved the manuscript. A few issues should still be addressed, as outlined below.

The title might still need some work, and by foregoing archaea and sticking to bacteria, perhaps the authors could consider or adapt the following: "investigating the relationship between CRISPR-Cas content and growth rate in bacteria".

Authors' response: We thank the reviewer for this constructive comment. It is really a more accurate title for our manuscript.

As the authors state in their rebuttal, they have divided bacteria into two groups according to growth rate. I think they should consider showcasing this split to display the data in figure 1 (possibly different colors in the same panel, or different panels) and add a bar graph visually showing growth rate between these two groups and determining statistical significance (besides table 2).

Authors' response: Yes. Five panels have been added in Fig. 1. As all the information in the previous Table 2 has now been presented in Fig. 1, we removed it from the revised manuscript.

Given the variation in cas gene number inherent to the CRISPR-Cas system nomenclature, the authors may elect to either keep the data and add a disclosure to that effect, or forego investigating cas gene number. Both may be agreeable, but the choice should be justified explicitly.

Authors' response: Frankly, we did not notice this phenomenon before. As the *Cas* gene number in each type of CRISPR-Cas system is relatively constant, the analysis of the *Cas* gene number in bacteria with only one particular CRISPR-Cas system type is meaningless. We removed them from analyzing bacteria with only class 1, class 2, type I, or type II CRISPR-Cas systems. However, when different types of CRISPR-Cas systems were analyzed together, the *Cas* gene number evolution could reflect the usage of different types of Cas gene clusters. Therefore, we retained most results on *Cas* gene number in the revised manuscript and described the reason in the MATERIALS AND METHODS section.

As the authors mention, temperature plays an important part in host biology. I also know that thermophilic organisms have been discussed to have "interesting" CRISPR-Cas patterns. The authors may elect to compare and contrast growth rate and CRISPR-Cas patterns between mesophiles and thermophiles, if and where applicable (a refined figure display, or options with different color usage).

Authors' response: Yes. Five panels have been added in Fig. 1.

As mentioned initially, the conflicting results from archaea are noteworthy and concerning. I suggest the authors focus on bacteria only in their study and then discuss archaea in the discussion section (but keep some of the data in the tables, perhaps not the figure), to contextualized this and also keep the most significant results and narrative in the paper. This is phylogenetically defensible, and if presented correctly, agreeable and understandable for the readership. "Only" bacteria are called out in the title afterall...

Authors' response: Very good comment. We were also worried about this subject. Now, the paragraph has been moved to the DISCUSSION section.

Figure 1 is a welcome addition, but needs improvements. The authors should provide several panels (see comment above), including bar graphs comparing and contrasting subsets of organisms and patterns (e.g. thermophilic vs mesophilic, fast growing vs slow growing, and possibly more). Visually, the authors should consider using a log scale for the y axis (log 10 scale might be appropriate given the numbers in play) and also a clearer label and scale for the x axis (not intuitive for microbiologists). Actual hours should be labelled, even if the scale is log-transformed. Also, the authors should consider adding relevant numbers (p-values) to the graph, and consider adding the R-square (on the figure, or in the legend, as the readership would assuredly wonder).

Authors' response: All the suggested revisions have been made.

Where applicable, statistical analyses must be performed to show significance (or not) of the data (e.g. Table 1, though tables 2 and 3 are noted).

Authors' response: The results of the statistical analysis of the data in Table 1 have been added in the note below the table.

Grammatically and stylistically, the paper would benefit from copy-editing, including for the table entries.

Authors' response: We checked the grammar and spelling of the final manuscript using the software Grammarly. Several stylistic errors in the tables have been corrected.

March 11, 2023

Dr. Deng-Ke Niu
Beijing Normal University
College of Life Sciences
Beijing 100875
China

Re: Spectrum03409-22R2 (Investigating the relationship between CRISPR-Cas content and growth rate in bacteria)

Dear Dr. Deng-Ke Niu:

Your manuscript has been accepted, and I am forwarding it to the ASM Journals Department for publication. You will be notified when your proofs are ready to be viewed.

Sincerely,

David Pride
Editor, Microbiology Spectrum
